# Investigation on the bond-slip behavior of recycled aggregate concrete-filled steel tube with studs

Gehao Cai[1], Bing Sun[1,2]*, Sheng Zeng[1], Peng Yang[3], Jie Zhang[1]

1 School of Civil Engineering, University of South China, Hengyang, China, 2 China Nuclear Construction Key Laboratory of High Performance Concrete, University of South China, Hengyang, China, 3 School of Civil and Architectural Engineering, Hainan University, Haikou, China

* sunbingnh@126.com

## Abstract

This study investigates the influence of built-in studs on the bond behavior in recycled aggregate concrete-filled steel tube (RACFST) composite structures through push-out experiments. The effects of stud number, position, and rows on RACFST bond strength and steel tube surface strain are systematically analyzed. Furthermore, the bond-slip behavior evolution mechanism is examined, and a constitutive equation is established. A novel interfacial modeling approach is developed via secondary development of ABAQUS software to comprehensively simulate RACFST interfacial bond-slip behavior. The results demonstrate that insufficient stud quantity compromises interface integrity, reducing bond strength, while increased stud count enhances composite stiffness and bond performance. Studs positioned nearer the free end extend the natural bond length participating in shear resistance, thereby improving bond strength. Internal studs promote stress redistribution within the composite structure, significantly improving collaborative performance. The proposed constitutive equation shows good agreement with experimental results, and the developed interface program accurately captures bond-slip curve trends. These findings facilitate RACFST applications and provide guidance for shear stud arrangement in RACFST structures.

## Introduction

The world remains in a period of extensive construction, accompanied by an increase in construction waste. Among these, the accumulation of waste concrete may cause environmental issues such as land waste and environmental pollution [1,2]. The effective reuse of waste concrete to achieve green and sustainable development of recycled concrete can significantly reduce its adverse impact on the environment. Compared with ordinary concrete, recycled aggregate concrete (RAC) has shortcomings such as a lower elastic modulus and higher drying shrinkage, which often limit its application in load-bearing structures [3]. To address the inherent defects of

**Data availability statement:** All relevant data are within the paper and its Supporting Information files.

**Funding:** The research is supported by the Hunan Provincial Natural Science Foundation (No.2025JJ90163).

**Competing interests:** The authors have declared that no competing interests exist.

RAC, researchers have proposed encasing recycled concrete in steel tubes, forming RACFST [4–6], a new type of composite structure that fully utilizes the advantages of steel tubes and RAC, thereby significantly enhancing the strength and ductility of core concrete columns through the confining effect of steel tubes [7]. The interfacial bonding performance serves as the foundation for this composite structure [8–10], directly influencing its mechanical properties and thus the safety of the component. When natural bonding fails to meet high force transfer demands, shear connectors like studs are introduced for internal force redistribution, ensuring structural performance amidst complex stress variations.

Over the past decades, research on the bond-slip behavior of concrete-filled steel tubes (CFST) conducted by scholars has been primarily focused on aspects such as steel types [11–13], concrete types [14–16], working environments [17–19]. Based on these factors, scholars explore characteristics such as the bond-slip mechanism [20], strength [21], constitutive relationships [22]. Shakir-Khalil [23] conducted a series of push-out tests on 40 short CFST specimens, and the results indicated that the ultimate push-out load is a function of the interfacial type, shape and size of the cross-section. Song et al. [24] compared the effects of different steel types, and found that the bond strength of a stainless-steel specimen was usually lower than that of a reference specimen with carbon steel at ambient temperature. Chen et al. [25,26] conducted research on the bond performance of CFST with patterned surfaces, concluding that the ultimate bond strength increased significantly with the increase in pattern height. Liu et al. [27] conducted repeated pushing experiments on recycled concrete-filled steel tube specimens with different concrete strengths and quality replacement rates of regenerated aggregate, and found that the ultimate bond strength increased with the increase of concrete strength. However, as the quality replacement rate of regenerated aggregate increases, the ultimate bond strength decreases. Lyu et al. [28] conducted a comparative study on 56 specimens, including both RACFST and CFST specimens. They found that the bond behavior of RACFST is similar to that of the steel-concrete interface in CFST and presented the empirical formula of bond-slip strength. Chen et al. [29,30] experimentally studied the bond-slip behavior of post-high-temperature circular CFST with RAC and high-strength concrete, finding that the bond strength of the RAC specimens initially decreased and then increased due to high temperature, while the high-strength concrete specimens exhibited the opposite trend. Sun et al. [31] through simulating a high chloride ion corrosive environment, concluded that bond performance exhibited a trend of first increasing and then decreasing as the corrosion rate increased. Amirreza et al. [32] carried out a study on lightweight concrete-filled steel tubes (LCFTs) with rock wool waste. They analyzed the bond strength and bond stress-slip behavior of LCFTs at high temperatures and developed a prediction model to describe how the bond strength varies with temperature. Tao et al. [33,34] studied the impact of post-fire on the bonding performance of CFST by simulating the high temperatures. In terms of the arrangement of shear connectors. Moreover, welding an inner ring on the inner surface of the steel tube is an effective method to improve the bonding performance. Chen et al. [35] conducted research on the shear performance of multi-row and

multi-column studs, proposing a calculation method for the shear capacity of stud groups. Dong et al. [36,37] studied the influence of studs and other types of connectors on the bond performance of CFST, pointing out that the performance-price ratio of studs and a circular rib for improving the bond strength is high, and the combination of studs and a circular rib can work well. Nevertheless, Roeder et al. [38] believed that shear connectors would reduce the bond strength of the interface. Yet, the bond slip properties of RACFST and CFST are very different. The mechanical properties of RACFST core concrete are worse than those of CFST, and it may be difficult to withstand the shear action brought by the studs, leading to premature damage of core concrete, consequently resulting in debonding, and thus affecting its normal working performance.

Research on the natural bond performance of CFST has been relatively comprehensive, covering aspects such as material properties and working environments. However, research on the bond-slip behavior of CFST with internal studs is still incomplete, and the enhancement effect of studs on the bond performance remains unclear. The complexity of the interaction between studs and concrete—combined with the uneven and asynchronous distribution of interfacial friction after stud installation—renders the bond-slip mechanism particularly challenging to characterize. To facilitate pipeline insertion and reasonable spatial allocation, the impact of the position of stud connectors on bond strength also needs to be investigated. Therefore, this article investigates the influence of the number of rings, the position of studs, and the number of stud rows on the bond-slip performance of RACFST specimens. It establishes the bond-slip evolution mechanism and constitutive equation specific to RACFST with internal studs. Furthermore, a novel connection interfacial setting program is developed, which can better simulate the bond-slip behavior of the interface during numerical simulations. This study clarifies the law of how the number and position of studs influence the bond performance of the RACFST structure. It provides a key reference for the design, has guiding significance for the installation and performance optimization of shear studs in this structure in practical engineering, and is conducive to promoting the widespread application and development of the RACFST structure in the construction field.

## Experimental design

### Specimen design

This study did not involve any human or animal participants conducted by the authors. No special permits were required for this research. The recycled coarse aggregate is the continuously graded recycled aggregate with a particle size of 0–20 mm, which is obtained after the waste concrete from a construction site in Hengyang is crushed, washed, graded and screened. The natural coarse aggregate is crushed stone with a continuously graded particle size of 0–20 mm. The fine aggregate has a fineness modulus of 2.4. P.O.42.5 grade Portland cement from Hengyang, Hunan Province was used. The design strength grade of this recycled concrete is C30. The weight mix proportion was cement: recycled aggregates: sand: water: water-reducing agent = 351: 1144: 726: 185: 0.817 kg m$^{-3}$. The physical performance indicators of aggregates were measured according to, as shown in Table 1 and Fig 1.

**Table 1. Physical properties of natural and recycled coarse aggregate.**

| Properties | Natural Coarse Aggregate | Recycled coarse aggregates |
| --- | --- | --- |
| Packing density (kg·m$^{-3}$) | 1447.6 | 1369.0 |
| Apparent density (kg·m$^{-3}$) | 2132.0 | 2532.2 |
| Water absorption (%) | 0.355 | 1.551 |
| Crushing index (%) | 8.614 | 14.937 |
| Clay content (%) | 0.0036 | 0.0314 |

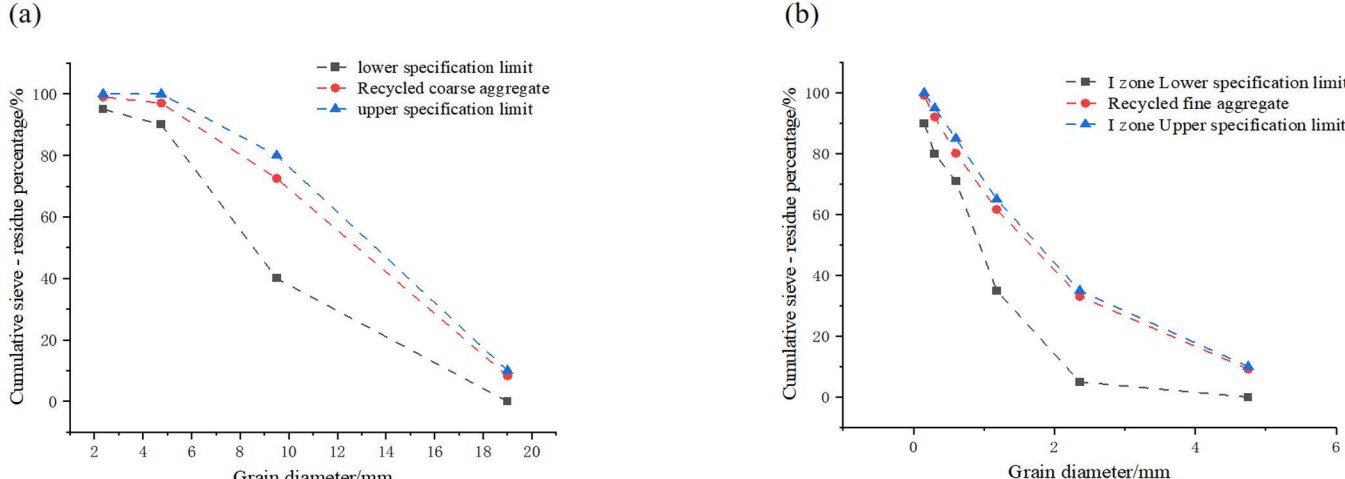

**Fig 1. Grading curves of coarse aggregates: (a) Recycled coarse aggregate; (b) Recycled fine aggregate.**

The RAC used in the experiment had a strength grade of C30, and the steel tubes were straight-welded Q345 tubes with an outer diameter of 168 mm and a thickness of 6 mm. The yield strengths of the steel tubes and studs are respectively 361 MPa and 350 MPa, while their tensile strengths are 558 MPa and 455 MPa, respectively.

Presently, the push-out test is a commonly used experimental method in academia to investigate the bond behavior of CFST [39,40]. The number of test specimens refers to References [26,41]. We designed seven RACFST specimens for push-out tests, with the variables being the position, rows and the number of rings of the studs. Specifically: RACFST-1 had no treatment applied to its inner wall; RACFST-2, RACFST-3, and RACFST-4 investigated the influence of stud positions (upper, middle, and lower) on the bond-slip behavior; RACFST-3, RACFST-5, and RACFST-6 explored the effect of the circumferential number of studs; RACFST-3 and RACFST-7 analyzed the impact of the number of stud rows.

The dimensions of the steel tubes and the specifications and placements of the studs are illustrated in Fig 2. The sizes and positions of the studs were referenced from literature [35,42]. Such stud setting includes the influence of stud position, the number of ring stud and the number of stud rows on the bond performance of RACFST, that is, the influence of various stud Settings on the bond slip performance of RACFST is comprehensively considered. To ensure a sufficient slip distance during the launching process, a 50 mm gap at the bottom end of the steel tube is intentionally left unfilled during concrete pouring. During the pouring of each RACFST specimen, three 150×150×150 mm cubic test blocks were simultaneously poured. Both were sealed with plastic film to strictly simulate the enclosed environment inside the steel tube, followed by 28 days of curing. the RACFST specimens were shown in Fig 3. On the day of the push-out test, the cubic test blocks were subjected to compressive strength testing to determine the core concrete strength of the RACFST. The compressive strength test is illustrated in Fig 4, and the test results are presented in Table 2.

## Test procedure

The experiment employed the YAW-J5000F micro-electro-hydraulic servo testing machine to load the specimen under displacement control. Push-out tests were conducted a push-out test at a rate of 1 mm min[-1] [43].The loading terminated when the displacement reached 20 mm. A steel block with a diameter slightly smaller than the inner diameter of the steel tube and a thickness of 50 mm was positioned at the loading end of the specimen, effectively transmitting the force from the testing machine to the core concrete and pushing it towards the free end. Strain gauges were adhered to the surface of the steel tube at distances of 20 mm, 50 mm, 110 mm, 185 mm, 260 mm, 335 mm, and 410 mm from the free end. A

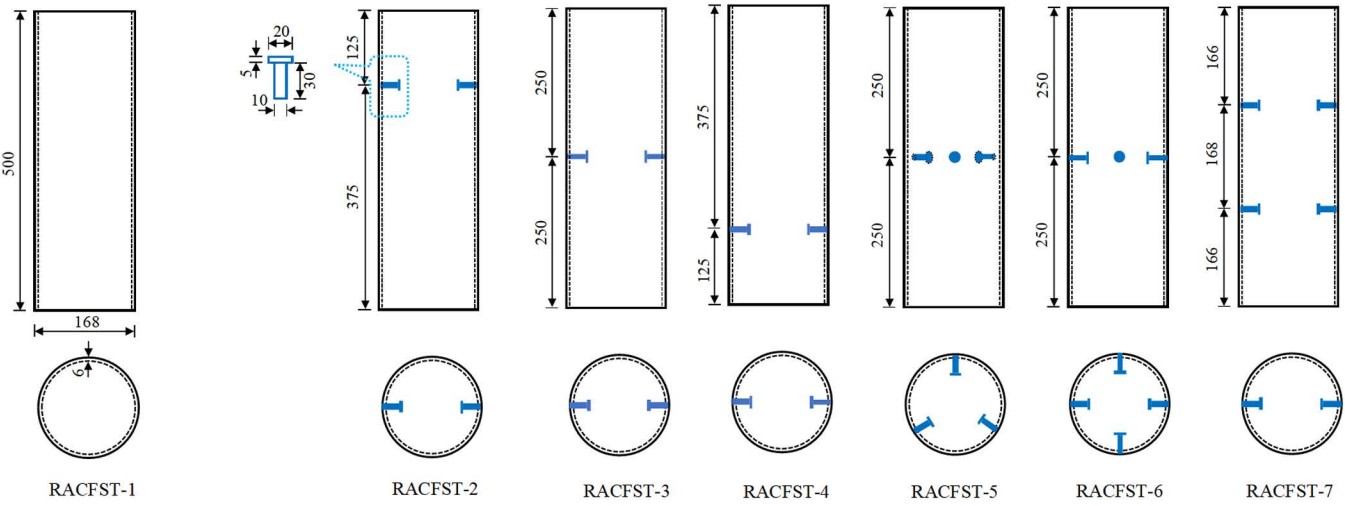

**Fig 2. Parameters of steel tubes and studs, stud positioning.**

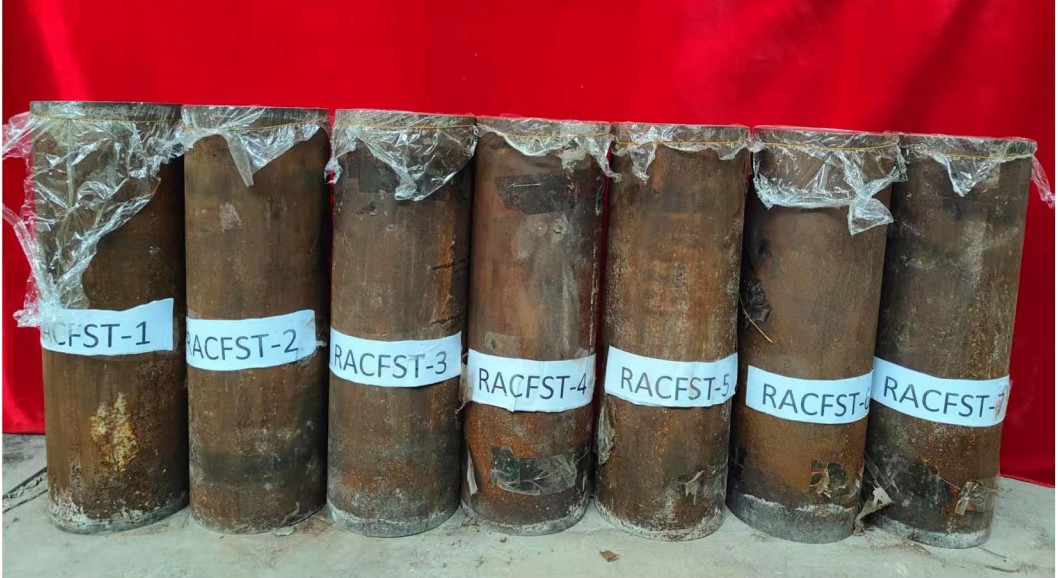

**Fig 3. Specimen of RACFST.**

uT7116Y static strain gauge was employed to measure the strain variation trends on the steel tube's surface during the push-out process of RACFST. The experimental loading and measurement setup is illustrated in Fig 5.

## Experimental results and discussion

### Test phenomena

During the initial phase of experimental loading, the adhesion between the steel tube and concrete interface remained excellent, with the concrete immediately adjacent to the studs undergoing continuous compression and densification,

**Fig 4. RAC compressive strength test.**

**Table 2. Compressive strength of core RAC.**

| Specimen | RACFST-1 | RACFST-2 | RACFST-3 | RACFST-4 | RACFST-5 | RACFST-6 | RACFST-7 |
|---|---|---|---|---|---|---|---|
| Compressive Strength(MPa) | 34.1 | 32.5 | 33.0 | 31.8 | 33.3 | 32.7 | 32.5 |

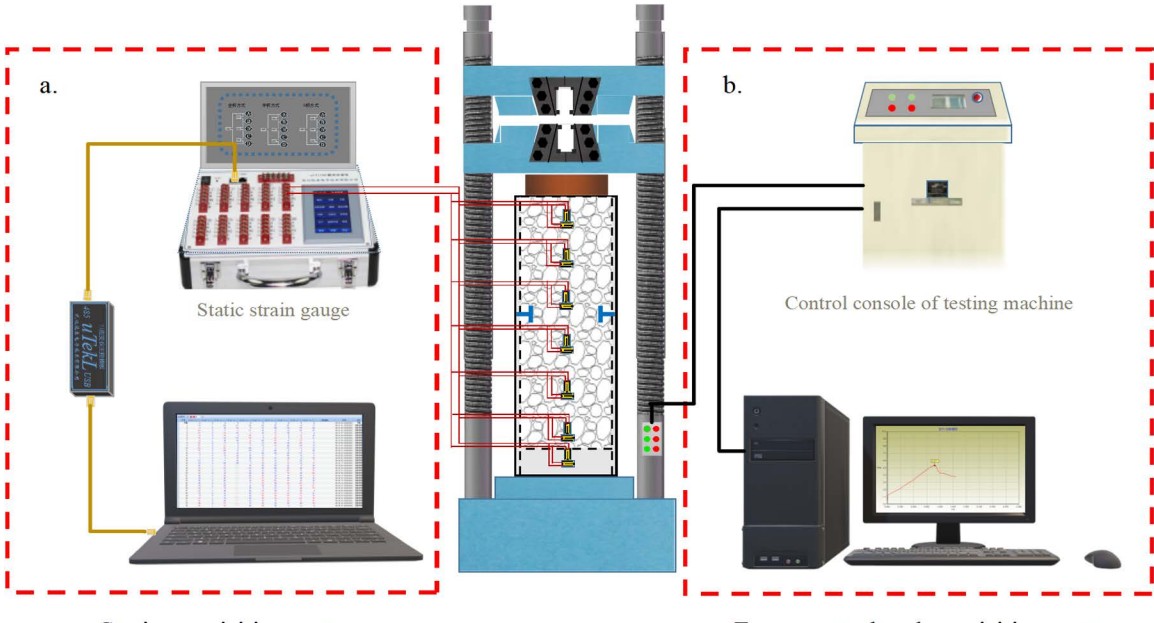

**Fig 5. Schematic diagram of test device.**

resulting in a linear increase in the slope of the load-slip curve. As the applied load increased, the slope of the curve gradually flattened, ultimately reaching a peak load. Immediately subsequent to this peak, there was a dramatic decrease in load, accompanied by distinct 'cracking' noises and the sound of concrete being crushed. In the later stages of loading, the load stabilized, remaining largely unchanged. Upon the conclusion of loading, the core concrete retained its structural integrity, with minimal discernible debris at the loading end. Close inspection revealed minute bulging on the outer wall of the steel tube near the studs, while the inner wall exhibited slight scratches indicative of concrete extrusion, as shown in Fig 6.

## Design parameter analysis

To investigate the impact of studs on the bonding effect, push-out tests were carried out on RACFST specimens with varying numbers, positions, and rows of studs. The measured load-slip curves are presented in Fig 7. The built-in studs influence the slip distance corresponding to the peak point. The slip corresponding to the peak bonding force of the specimen in the natural bonding state (RACFST-1) is approximately 2 mm. Before attaining the peak load, the applied load rises rapidly with the slip distance. However, the slip corresponding to the peak point of the specimens with built-in studs is approximately 8 mm. Within a slip of 2 mm, the pattern is consistent with that of RACFST-1. However, between 2 mm and 8 mm, the specimens with built-in studs have an additional stud-resistance stage. The load-slip curve of this stage is smoother than that of the specimens without studs before reaching the peak load. Overall, the different arrangements of studs not only modify the numerical value of the bonding bearing capacity but also exert a certain influence on the overall variation tendency of the load-slip curve.

From Fig 7a, it can be observed that the bonding force of the natural bonding specimen is higher than that of the specimen with two studs when only one row of studs is present. As the number of circumferential studs increases from

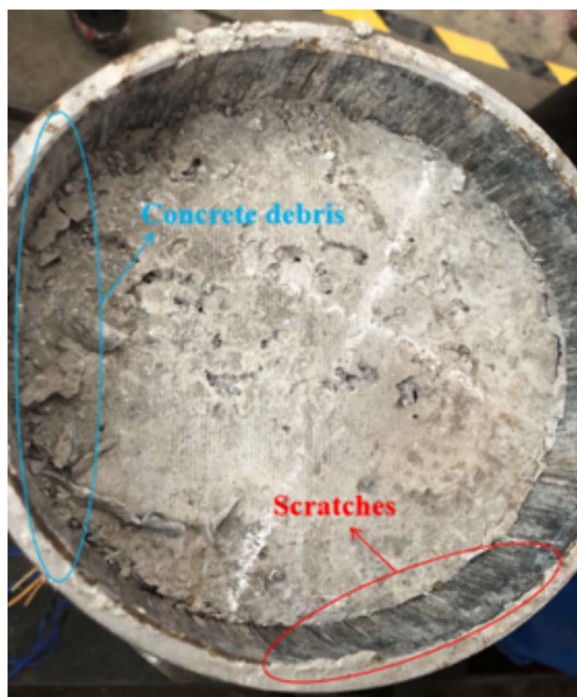

**Fig 6. Concrete debris and scratches.**

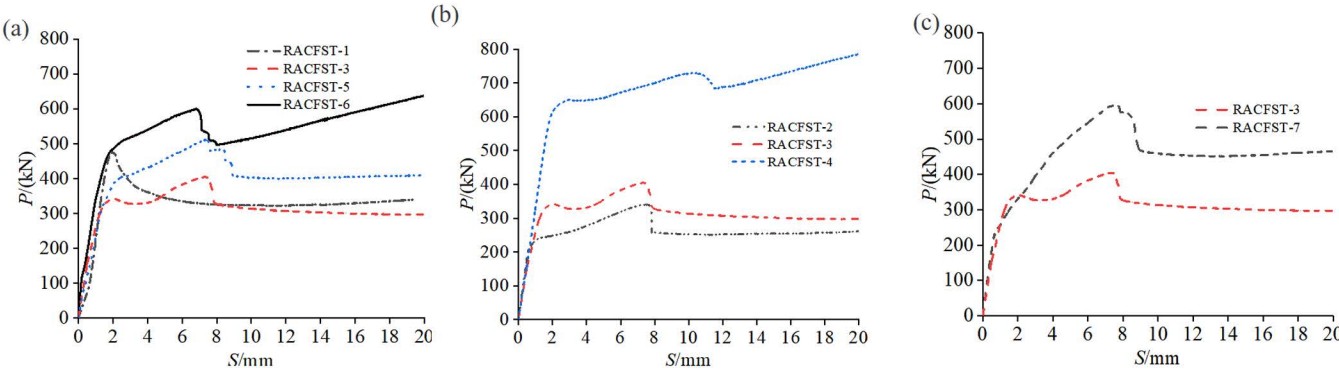

**Fig 7. Load-slip curves for different design parameter specimen: (a) Series of stud numbers; (b) Series of stud positions; (c) Series of stud rows.**

2 to 3 and then to 4, the ultimate bonding force increases by 42.48% and 76.70%, respectively, when compared to the specimen with two studs. Fig 7b shows that the closer the stud position is to the free end, the larger the ultimate bonding force becomes, and this enhancement effect is more pronounced as the stud moves closer to the free end. Furthermore, Fig 7c indicates that when the number of stud rows changes from single to double, there is a notable increase in the peak bonding force by 47.5%.

When the number of studs is small, the push-out operation will induce shear deformation in the studs, leading to failure at a diffusion angle based on the principle of force diffusion. This undermines the original natural bonding between the steel tube and the RAC, weakening the integrity of the bonding interface.Consequently, the natural bonding portion of the RAC with the steel tube, where no slip has occurred yet, fails, resulting in a decrease in the bonding strength. However, as the number of studs increases, the combined stiffness of the RAC and studs also increases, which outweighs the potential damage to the interface integrity caused by the studs, thereby enhancing the bonding force. Independently analyzing the impact of the number of studs on the bonding force reveals that the greater the number of studs, the stronger the interface bonding force becomes. Moreover, the position of the studs exerts a significant influence on the bonding performance. Specifically, the closer the studs are to the free end, the greater the ultimate bonding force achieved. The shear resistance capacity of the RACFST interface with built-in studs is primarily determined by two factors: the natural bonding between the steel tube and the RAC, and the shear resistance capacity of the studs themselves. The distance from the stud to the loading end, which represents the length of the natural bonding between the steel tube and the RAC that contributes to the shear resistance. With a constant stud count, lower stud positions increase this effective bond length, raising both peak bonding force and displacement. Furthermore, the greater the number of stud rows, the stronger the interface bonding force becomes. The addition of double rows of studs further increases the combined stiffness of the interface, thereby enhancing the ultimate bonding performance, and the group stud effect can also improve its bond resistance. These conclusions can provide a reference for adjusting the spatial positioning of shear connectors in composite structures and the rational distribution of bond strength.

## Strain analysis

The strain distribution on the surface of the steel tube during the push-out process reflects the transfer effect of the bonding force between the RAC and the steel tube, namely the cooperative working performance of the two [44].

The comparison of the longitudinal strain distribution on the surface of the steel tube of the natural bonding specimen RACFST-1 and the typical specimen RACFST-5 with built-in studs is depicted in Fig 8. On the whole, the longitudinal

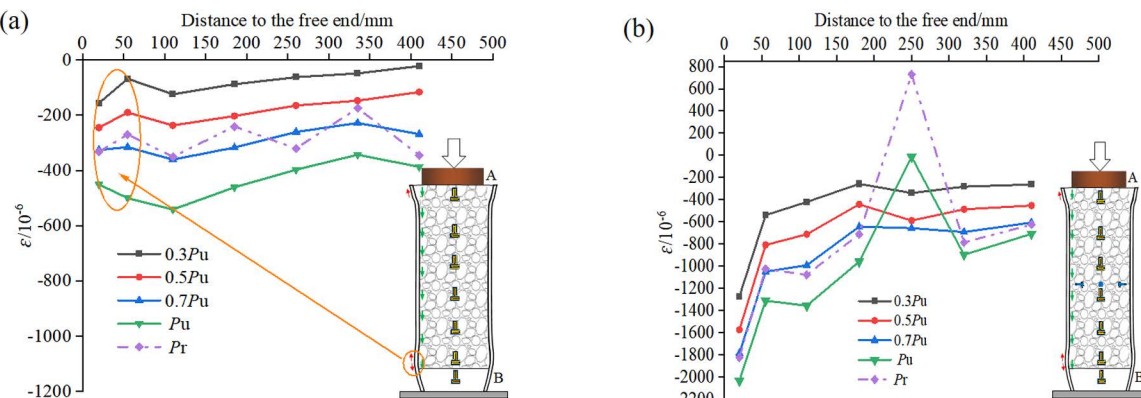

**Fig 8. Longitudinal strain distribution on the steel tube surface of the specimen: (a) Natural bonding specimen RACFST-1; (b) Built-in studs specimen RACFST-5.**

strain distribution on the surface of the steel tube escalates progressively with the increase of the load gradient. The residual load Pr is smaller than the peak load Pu, thus the corresponding surface strain is also lower than the peak strain. Under the same load gradient, from the loading end (upper end) to the free end (lower end) of the steel tube, the strain exhibits an ascending tendency. In Fig 8a, there are also some exceptions close to the free end. This is because there is a "strain distortion" area at the free end and there is a certain tensile strain, leading to a certain reduction in compressive strain. In Fig 8b, studs are arranged at a height of 250 mm of the steel tube of specimen RACFST-5. When the load is small (less than $0.8P_u$), the effect of the studs is not conspicuous. But after the load reaches Pu, the effect of the studs is exerted, and the strain on the surface of the steel tube undergoes a local mutation, with the compressive strain decreasing or even transforming into tensile strain.

The transverse strain on the surface of the steel tube reflects the lateral hoop force acting on the concrete during the push-out process [45]. The comparison of the transverse strain distribution on the surface of the steel tube of the natural

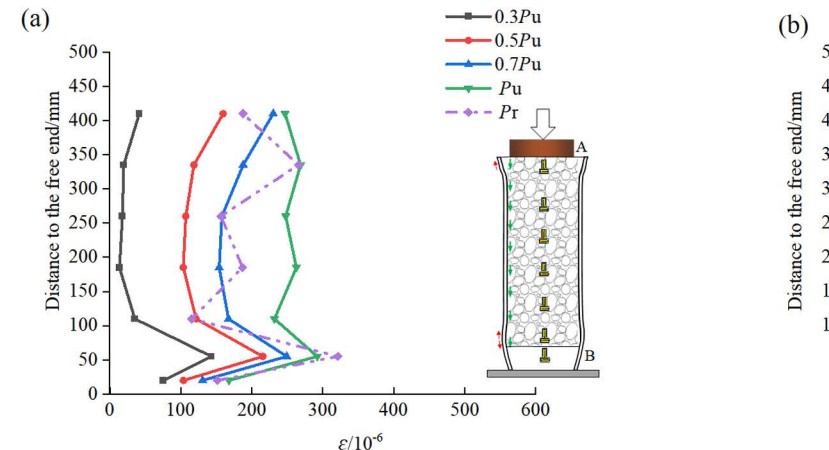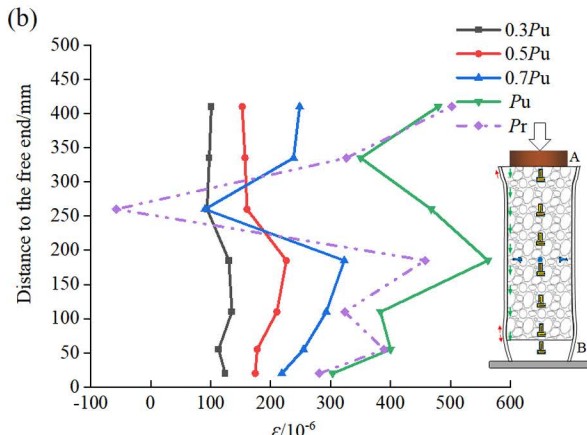

**Fig 9. Transverse strain distribution on the steel tube surface of the specimen: (a) Natural bonding specimen RACFST-1; (b) Built-in studs specimen RACFST-5.**

bonding specimen RACFST-1 and the typical specimen RACFST-5 with built-in studs is presented in Fig 9. Before attaining the peak load $P_u$, the strain growth at each measuring point is relatively stable.

Nevertheless, their change trends are discrepant after the peak load Pu. As can be observed from Fig 9a, a sudden change in strain emerges near the free end, which might be related to the "wedging effect" of the free end. As the concrete gradually wedges into the hollow steel tube, the hoop restraint on the concrete increases conspicuously. In Fig 9b, when the load is less than 0.5Pu, the effect of the studs is negligible. But thereafter, near the location where the studs are set, a stress mutation occurs on the surface of the steel tube. Due to the studs being compressed and bent, under the influence of the load, the studs impart an inward retraction force to the steel tube, leading to the transverse strain on the surface of the steel tube being smaller than that of the natural bonding specimen.

Upon comparing the surface strains of the steel tubes in the specimens, it is observed that the local positions where the studs are situated redistribute the relatively high strains originally endured by the specimens. This alleviates the strain borne solely by the steel tube and instead distributes it jointly among the steel tube, concrete, and studs. Additionally, with the inclusion of the built-in studs, under all loading levels, the strains experienced by the entire length of the steel tube (excluding the areas influenced by the studs) surpass those of the naturally bonded specimen. This suggests that the strains in the RAC are effectively transferred to the surface of the steel tube. Consequently, the overall performance and cooperative working capability of the composite structure are significantly and effectively enhanced after the installation of studs.

## Bonding-slip evolution mechanism analysis

Based on the interfacial bonding-slip curves, the curve constitutive model of the RACTST with built-in studs was summarized and induced, as presented in Fig 10. The model is categorized into four stages: the concrete compaction stage (OA), the concrete and stud interaction stage (AB), the concrete/stud crushing stage (BC), and the stable slip stage (CD). The bonding-slip failure mechanism of the RACFST interface with built-in studs is depicted in Fig 11.

Concrete compaction stage (OA): In the early stage of loading, the interface is in the elastic phase, and the slip amount is relatively small. The concrete is compressed and densified, and then begins to have elastic contact with the studs.The interfacial bonding is primarily attributed to the chemical bonding force stemming from the natural adhesion between the steel tube atop the studs and the RAC, mechanical interlocking forces resulting from the rough surface textures of the

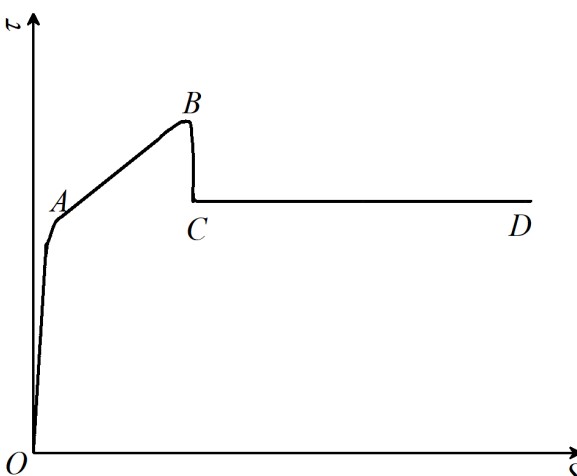

**Fig 10.** $\tau$-*s* constitutive mode.

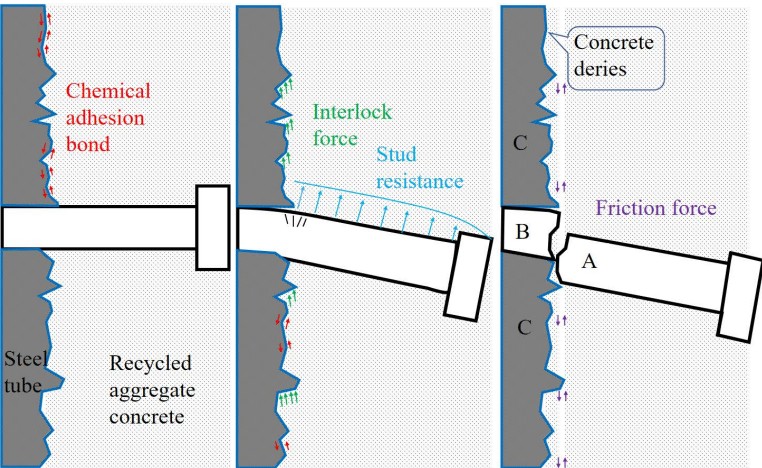

**Fig 11. Failure mechanism of bond-slip.**

steel tube and RAC, as well as the mutual elastic compression occurring between the studs and RAC. At this stage, the concrete is compressed and micro-cracks form [46].

Concrete and stud interaction stage (AB): The RAC is in tight contact with the studs, creating a region at the interface with a diffusion angle of approximately 30° [47]. As the concrete and studs interact with each other, the effect of the studs gradually intensifies and becomes dominant. Until the studs are sheared off or the concrete is crushed, the bonding force at this stage is primarily provided by the resistance between the studs and the concrete.

Concrete/stud crushing stage (BC): The Interfacial bonding approaches or reaches the combined limit of the ultimate bearing capacity of both the concrete and the studs, as well as the natural bonding force. At this stage, the concrete within the stud failure zone undergoes rapid and localized crushing, or the studs are severed, resulting in a stepped and sudden drop in the bonding force.

Stable slip stage (CD): The concrete above the studs is continuously crushed or the studs are sheared, and the interface is fully penetrated, enabling the bonding force to develop steadily. The bonding force is jointly offered by the chemical bonding force, the mechanical interlocking force, the friction force of the sheared studs embedded in the concrete that stably acts on the steel tube, or the continuous crushing force of the concrete. The sum of the three is stable. Additionally, the change trend of the bonding force in the stable slip section is also determined by the macroscopic deviation of the steel tube. Owing to the non – uniform of the steel tube, the curve may exhibit three development trends: rising, falling, and remaining constant. For instance, the stable slip section of RACFST-4 shows an upward trend.

## Numerical simulation of bonding-slip

The core principle of the ABAQUS finite element method is to divide a complex system into manageable parts, assemble and analyze them. This approach enables precise mechanical simulations of deformable bodies with intricate geometries under the influence of diverse external forces [48,49]. The bonding-slip phenomenon at the RACFST contact interface embodies a complex mechanical behavior, involving intricate interactions between the materials and multiple loads. While experiments primarily explore its performance changes from a macroscopic perspective, to visually represent its stress distribution, cloud maps, and other details, we need to supplement these insights with finite element simulations. However, the existing simulations of the interface bonding-slip cannot accurately and comprehensively simulate the entire bonding-slip process. Therefore, we conducted in-depth research.

## Constitutive material model

**Model of steel tube.** The steel tube employs the C3D8R solid element. Equation (1) is the five-stage stress-strain plastic analysis model utilized for the steel tube [50].

$$\sigma = \begin{cases} E_s\varepsilon & (\varepsilon \le \varepsilon_e) \\ -A\varepsilon^2 + B\varepsilon + C & (\varepsilon_e < \varepsilon \le \varepsilon_{e1}) \\ f_y & (\varepsilon_{e1} < \varepsilon \le \varepsilon_{e2}) \\ f_y\left[1 + 0.6\frac{\varepsilon - \varepsilon_{e2}}{\varepsilon_{e3} - \varepsilon_{e2}}\right] & (\varepsilon_{e2} < \varepsilon \le \varepsilon_{e3}) \\ 1.6f_y & (\varepsilon > \varepsilon_{e3}) \end{cases}$$

(1)

Where $f_p$, $f_u$ and $f_y$ represent the proportional limit, tensile strength and yield strength limit of the steel respectively; $\varepsilon_e$, $\varepsilon_{e1}$, $\varepsilon_{e2}$ and $\varepsilon_{e3}$ are the strains corresponding to the proportional limit, the onset of the yield stage, the onset of the strengthening stage and the ultimate tensile strength respectively, $\varepsilon_e = 0.8f_y/E_s$, $\varepsilon_{e1} = 1.5\varepsilon_e$, $\varepsilon_{e2} = 10\varepsilon_{e1}$, $\varepsilon_{e3} = 100\varepsilon_{e1}$; $E_s$ is the elastic modulus of the steel, $E_s = 2.06 \times 10^5$MPa; $A = 0.2f_y/(\varepsilon_{e1}-\varepsilon_e)^2$, $B = 2A\varepsilon_{e1}$, $C = 0.8f_y + A\varepsilon_e^2 - B\varepsilon_e$.

**Model of RAC.** The concrete in RACFST pertains to confined concrete, which differs from the constitutive model of ordinary uniaxial compressive concrete. Hence, the stress-strain constitutive model of confined concrete in the literature [51] is chosen, as presented in Equation (2). The concrete plastic damage is defined to simulate the nonlinear behavior of RAC. The dilation angle, eccentricity, ratio of biaxial to uniaxial compressive strength of concrete, influence parameters of the concrete yield form and the concrete viscosity coefficient are 30°, 0.1, 1.16, 0.6667 and 0.0005 respectively.

$$y = \begin{cases} 2x - x^2, & x \le 1 \\ \frac{x}{\beta(x-1)^2 + x}, & x > 1 \end{cases}$$

(2)

where $x = \varepsilon/\varepsilon_0$; $y = \sigma/\sigma_0$; $\sigma_0 = f_c$; $\xi = A_sf_y/A_cf_{ck}$; $\varepsilon_0 = \varepsilon_c + 800 \times \xi^{0.2} \times 10^{-6}$; $\varepsilon_c = (1300 + 12.5 \times f_c) \times 10^{-6}$; $\beta = (2.36 \times 10^{-5})^{[0.25+(\xi-0.5)^7]}f_c^{0.5} \times 0.5 \ge 0.12$; $f_c$ represents the compressive strength of concrete cylinders.

**Model of bonding interface.** The definition of interface contact is crucial for the simulation of bonding-slip behavior in RACFST. When the interface slips, the bonding force primarily comprises three components: normal, transverse tangential, and longitudinal tangential. The normal and transverse tangential deformations are considerably smaller than the longitudinal tangential deformation. Therefore, the longitudinal tangential direction is considered the primary research focus for bonding-slip behavior [52,53]. Existing simulations of interfacial bonding-slip have mainly employed two approaches: setting coulomb friction [54], or using nonlinear spring elements [55]. Coulomb friction is calculated as the product of the interfacial friction coefficient and the normal pressure. However, it can only simulate a linear change process. Since bonding-slip is a complex nonlinear phenomenon, using Coulomb friction alone is unable to accurately simulate the tangential force transfer behavior of the interface. The simulation of nonlinear spring elements is achieved by establishing connection springs between the nearest meshes of the steel tube and concrete in the longitudinal tangential direction. The nonlinear constitutive behavior of the spring elements is precisely defined using bonding-slip parameters measured in experiments, allowing for a relatively precise simulation of the bonding interface. However, the setup of the spring elements can be excessively complicated. Therefore, in this paper, based on the bond-slip concept for "point" spring elements, we have simplified and extend this approach to the "surface" of the connection interface. A new method for setting the connection interface using a python script is proposed, this method effectively remedies the shortcoming of previous methods, which were unable to simulate the descending segment of the bond-slip curve, and achieves rapid and precise simulation of the bond-slip performance of RACFST interfaces equipped with internal studs. The setup process of this new method for configuring the connection interface is illustrated in Fig 12.

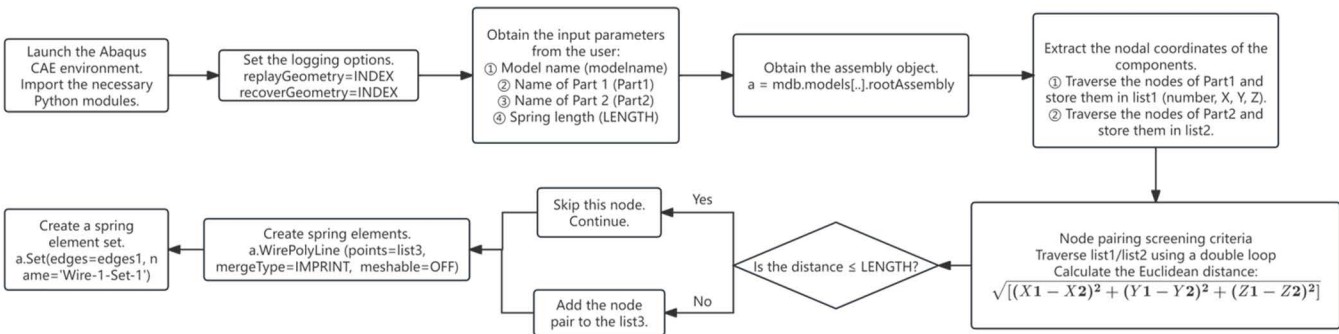

**Fig 12. Flow chat of the new method for setting the connection interface.**

The main concept of establishing the connection interface with the Python script is to utilize the script to create the connection interface, define the elastic nonlinear behavior in the ABAQUS "Interaction-connector section" module, and input the nonlinear parameters of the bond-slip to precisely simulate the bond-slip behavior of the interface. This paper mainly employs this method to establish the bonding interface. The specific establishment steps are as follows:

Step one: Input the python script, and then enter follow the prompts to enter Model-1, steel-1, concrete-1, and the distance judgment conditions in order to create a connection area.

Step two: Select the nonlinear behavior of the connection interface and input the bond-slip constitutive parameters.

Step three: Assign the appropriate direction, rotation angle, and rotation axis to the connection interface, as shown in Fig 13, to complete the establishment of the connection section.

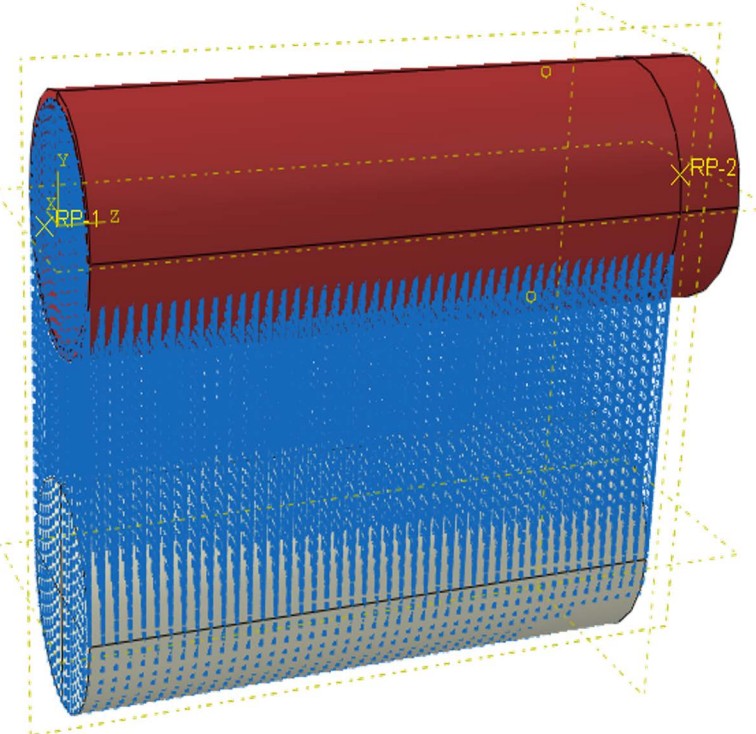

**Fig 13. Bond interface between steel tube and concrete.**

**Model of studs.** The commonly employed shear connectors are studs welded on the inner wall of the steel tube, with the studs embedded in the concrete and bonded to the steel tube. Equation (3) is the three-linear constitutive model utilized for the studs [56,57].

$$\sigma_s = \begin{cases} \varepsilon_s \cdot E_s, & 0 \leq \varepsilon_s < \varepsilon_{ys} \\ \sigma_{ys} + \frac{\sigma_{us}-\sigma_{ys}}{\varepsilon_{us}-\varepsilon_{ys}}\left(\varepsilon_s - \varepsilon_{ys}\right), & \varepsilon_{ys} \leq \varepsilon_s \leq \varepsilon_{us} \\ \sigma_{us}, & \varepsilon_{us} < \varepsilon_s \end{cases} \tag{3}$$

Where $\sigma_{ys}$, $\sigma_{us}$, $\varepsilon_{ys}$ and $\varepsilon_{ys}$ represent the yield strength, ultimate strength, yield strain and ultimate strain of the studs respectively, and $E_s$ is the nominal elastic modulus of the studs. The yield strain and ultimate strain are respectively taken as 0.2% and 0.6%.

## Comparison between experimental and simulation results

The numerical simulation analysis was conducted on the specimens RACFST-2 to RACFST-7 with built-in studs. The comparison between the experimental and simulation results is shown in Fig 14. It can be observed that the overall change trends of the simulation and experimental curves are consistent, including the rising section, peak point, falling section and stable section. By comparing the experimental data with the ultimate bond strength results from the finite element model and the proposed constitutive model, we found that the finite element model maintains average errors within 7%, effectively simulating the entire bond-slip process. This satisfactorily addresses the issue that the traditional method is unable to simulate the process of the bonding force falling section [35].

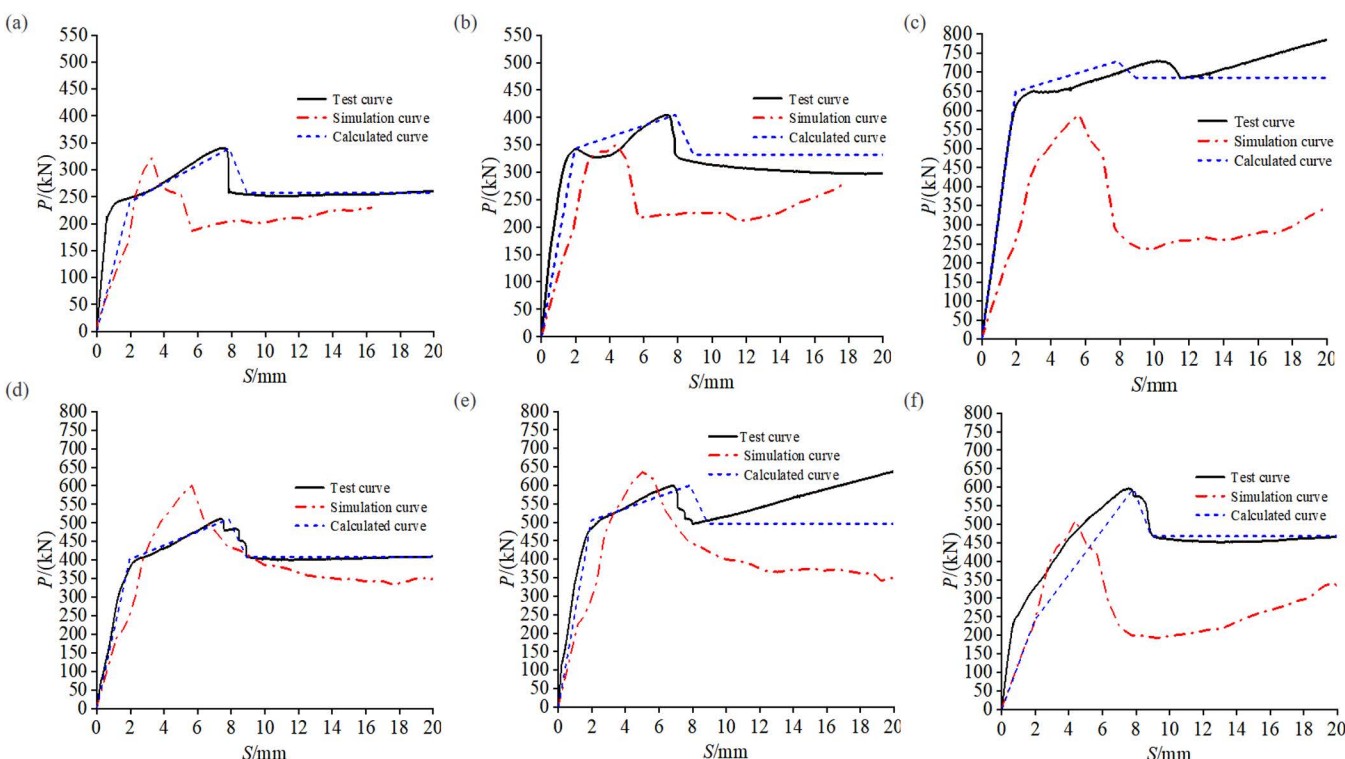

**Fig 14. Results of individual built-in studs specimens from tests, simulations and calculations: (a) RACFST-2; (b) RACFST-3; (c) RACFST-4; (d) RACFST-5; (e) RACFST-6; (f) RACFST-7.**

The author selected the optimal mesh size through sensitivity analysis. Mesh sizes of 5, 10, 15, and 20 millimeters were considered. Through the operation, it was found that although the difference between the 5-mm and 10-mm mesh sizes was negligible, there was still an observable difference. Therefore, 10 millimeters was chosen as the mesh size.

In the curves of Fig 14, the values of the characteristic points of the experimental and simulation results for each specimen are close, but the deviation between the simulation and experimental results of RACFST-4 is relatively large. This is mainly due to the fact that the later stable slip section of RACFST-4 in the experiment is influenced by macroscopic deviations, while this factor is not taken into account in the numerical simulation. Hence, the simulation results do not present a rising section similar to that of the experiment. The analysis emphasis of this paper is to verify the applicability of the proposed new method. The consistent values of the characteristic points and the curve change trends of the simulation suggest that this method can effectively simulate the entire process of the bond-slip performance of RACFST with built-in studs. In the subsequent stage, upon acquiring obtaining a substantial amount of experimental data, the accuracy of the model can be further adjusted and the parameter analysis can be expanded to enhance the applicability of the finite element method.

In some specific cases, such as the stable slip stage of RACFST-4, significant deviations occur. The large deviation in the stable slip stage of RACFST-4 stems from macroscopic irregularities, such as the irregular roundness of the steel tube or the inconsistent dimensions at its two ends. These irregularities, which are inevitable during the steel tube manufacturing process, may cause the curve to show three development trends: rising, falling, and stabilizing. As a result, this macroscopic deviation occurs and has a decisive impact on the development of the load – slip curve and the sliding performance of the interfacial combination [45].

## Mises stress distribution

As depicted in Fig 15, the distribution of Mises stress in RACFST is presented under both natural bond conditions and the influence of studs. In the naturally bonded specimens, it is evident that the Mises stress gradually increases along the steel tube surface from the loading end towards the free end. The maximum bond force occurs at the free end of the steel tube, whereas in the concrete, it is situated at the concrete loading end. Upon the installation of studs, a notable change in the overall Mises stress distribution along the specimen's height becomes evident. The studs, subjected to substantial

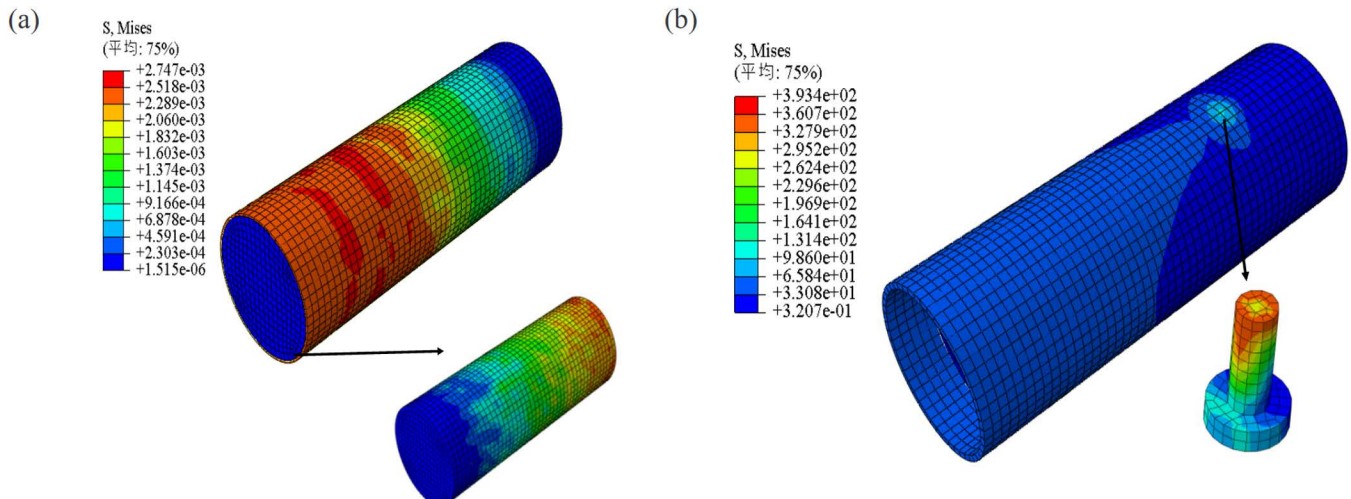

**Fig 15. Mises stress distribution in RACFST under two conditions: (a) natural bond conditions; (b) Internal stud conditions.**

loads, effectively alleviate the bond force borne by the steel tube, resulting in the formation of a diffusion zone along the steel tube's surface, spanning approximately a 30° angle. The ends of the studs are subjected to considerable Mises stress.

The results indicate that after the installation of studs, the distribution of Mises stress along the height of the specimen undergoes significant changes. The studs bear a substantial portion of the load, effectively reducing the bonding force endured by the steel tube. This implies that the studs alter the internal force transfer mechanism of the structure, leading to a more uniform stress distribution, which positively enhances the overall load-bearing capacity and stability of the structure.

## The calculation method for bond Strength

**Characteristic value of bonding strength.** The definition of bond strength at characteristic points, denoted as $P_i$ (comprising initial bond strength $P_s$, peak bond strength $P_u$, and residual bond strength $P_r$), corresponds respectively to the characteristic bond strengths $\tau_i$ (initial bond strength $\tau_s$, peak bond strength $\tau_u$, and residual bond strength $\tau_r$) and characteristic slip displacements $S_i$ (initial slip $S_s$, peak slip $S_u$, and residual slip $S_r$). The values of these characteristic parameters for the tested specimens, as obtained from experiments, are presented in Table 3. As evident from Table 3, the experimental average values for $S_s$, $S_u$, and $S_r$ are 1.96 mm, 7.85 mm, and 8.90 mm, respectively. Given the intricate interplay of various uncertain factors influencing the stud slip displacement, such as the interfacial friction coefficient, macroscopic deviations, as well as the complexities inherent in the studs, concrete, and steel tubes, it is prudent to adopt the experimental averages of the calculated characteristic slip values for the studs embedded within as the basis for the constitutive model fitting analysis. Its report statistical measures are shown in Table 4.

The calculated values of $P_s$, $P_u$ and $P_r$ have little difference from the experimental values, and the root-mean-square error (RMSE) and standard deviation are also small, indicating that the calculated values are close to the experimental values, the calculation accuracy is high, and the relevant calculation model or method may be relatively reliable. The proposed constitutive model demonstrates extremely small errors (within 0.1%) in predicting the ultimate bond strength ($P_u$) when compared with the experimental data. However, $P_s$ and $P_r$ are more discrete than $P_u$, indicating that the accuracy of the calculation model needs to be further optimized.

**Table 3. Specimen Characteristics.**

| Specimen | Experimental values | | | | | | Calculated values | | | | | |
|---|---|---|---|---|---|---|---|---|---|---|---|---|
| | $P_s$ (kN) | $S_s$ (mm) | $P_u$ (kN) | $S_u$ (mm) | $P_r$ (kN) | $S_r$ (mm) | $P_s$ (kN) | $S_s$ (mm) | $P_u$ (kN) | $S_u$ (mm) | $P_r$ (kN) | $S_r$ (mm) |
| RACFST-1 | 376.0 | 1.42 | 475.6 | 1.95 | 328.6 | 7.13 | – | – | – | – | – | – |
| RACFST-2 | 240.4 | 1.24 | 339.7 | 7.55 | 257.4 | 7.83 | 240.4 | 1.96 | 339.7 | 7.85 | 257.4 | 8.90 |
| RACFST-3 | 342.8 | 2.03 | 404.2 | 7.35 | 331.4 | 7.87 | 343.0 | 1.96 | 404.3 | 7.85 | 331.6 | 8.90 |
| RACFST-4 | 648.4 | 2.84 | 728.3 | 10.43 | 684.2 | 11.50 | 649.2 | 1.96 | 728.8 | 7.85 | 685.6 | 8.90 |
| RACFST-5 | 401.7 | 2.41 | 510.5 | 7.38 | 407.6 | 9.05 | 401.7 | 1.96 | 510.5 | 7.85 | 407.6 | 8.90 |
| RACFST-6 | 504.6 | 2.45 | 599.6 | 6.79 | 496.0 | 8.09 | 504.6 | 1.96 | 599.6 | 7.85 | 496.0 | 8.90 |
| RACFST-7 | 243.9 | 0.78 | 596.3 | 7.59 | 467.1 | 9.06 | 241.0 | 1.96 | 596.3 | 7.85 | 467.4 | 8.90 |

**Table 4. Reported statistical methods of Characteristic points of bond-slip.**

| | Average deviation | RMSE | Standard deviation |
|---|---|---|---|
| $P_s$ | −0.317 | 1.231 | 1.331 |
| $P_u$ | 0.100 | 0.207 | 0.204 |
| $P_r$ | 0.317 | 0.590 | 0.564 |

Based on the experimental results, a statistical regression analysis was performed on the characteristic values of bond strength for RACFST incorporating studs. This analysis considered variables such as the number of studs, number of rows, area, tensile strength, and natural bond length. Given that the positioning of studs is influenced by their natural bond length, a statistical regression formula incorporating this factor was incorporated into the model. As a result, the formulas for calculating the characteristic values of bond strength, taking into account these three key factors, follow as Equation (4)-(6), respectively.

Considering the number of studs.

$$\begin{cases} P_s = 263.774 + 25.929e^{\frac{nA_sf_u}{1.482A}} \\ P_u = 1059.157 - 933.243e^{-\frac{nA_sf_u}{4.659A}} \\ P_r = -141.166 + 350.502e^{\frac{nA_sf_u}{5.521A}} \end{cases}$$

(4)

Considering the position of studs.

$$\begin{cases} P_s = 188.735 + 17.304e^{0.00828\pi Rl_x\tau_{u1}} \\ P_u = 323.608 + 3.200e^{0.01768\pi Rl_x\tau_{r1}} \\ P_r = 237.747 + 4.125e^{0.01712\pi Rl_x\tau_{r1}} \end{cases}$$

(5)

Considering the row of studs.

$$\begin{cases} P_s = 0.351m\left(263.774 + 25.929e^{\frac{nA_sf_u}{1.482A}}\right) \\ P_u = 0.738m\left(1059.157 - 933.243e^{-\frac{nA_sf_u}{4.659A}}\right) \\ P_r = 0.705m\left(-141.166 + 350.502e^{\frac{nA_sf_u}{5.521A}}\right) \end{cases}$$

(6)

Where $A_s$ represents the cross-sectional area of the stud shank in mm²; $f_u$ is the tensile strength of the stud in MPa; $R$ denotes the diameter of the stud in mm; $l_x$ signifies the distance from the stud position to the loading end in mm; $A$ stands for the effective contact area between the steel tube and the concrete in mm²; and $m$ represents the number of rows of studs ($m \geq 2$).

**Bonding-slip constitutive equation**

The bond strength at characteristic points is calculated using Equation (7).

$$\tau_i = P/A$$

(7)

Where $\tau_i$ refers to the bond strength at the characteristic point, measured in MPa, while $P$ represents the bond force value at that characteristic point, expressed in kN.

Based on the segmented fitting of the bond-slip strength values at various characteristic points, the τ-S curve expression is derived as shown in Equation (8). Fig 14 illustrates the excellent correlation between the theoretical calculations and experimental results, confirming their consistency.

$$\tau = \begin{cases} \frac{\tau_s}{S_s}S, & 0 \leq S \leq S_s \\ \tau_s + \frac{S-S_s}{S_u-S_s}(\tau_u - \tau_s), & S_s \leq S \leq S_u \\ \tau_u - 0.422, & S_u \leq S \end{cases}$$

(8)

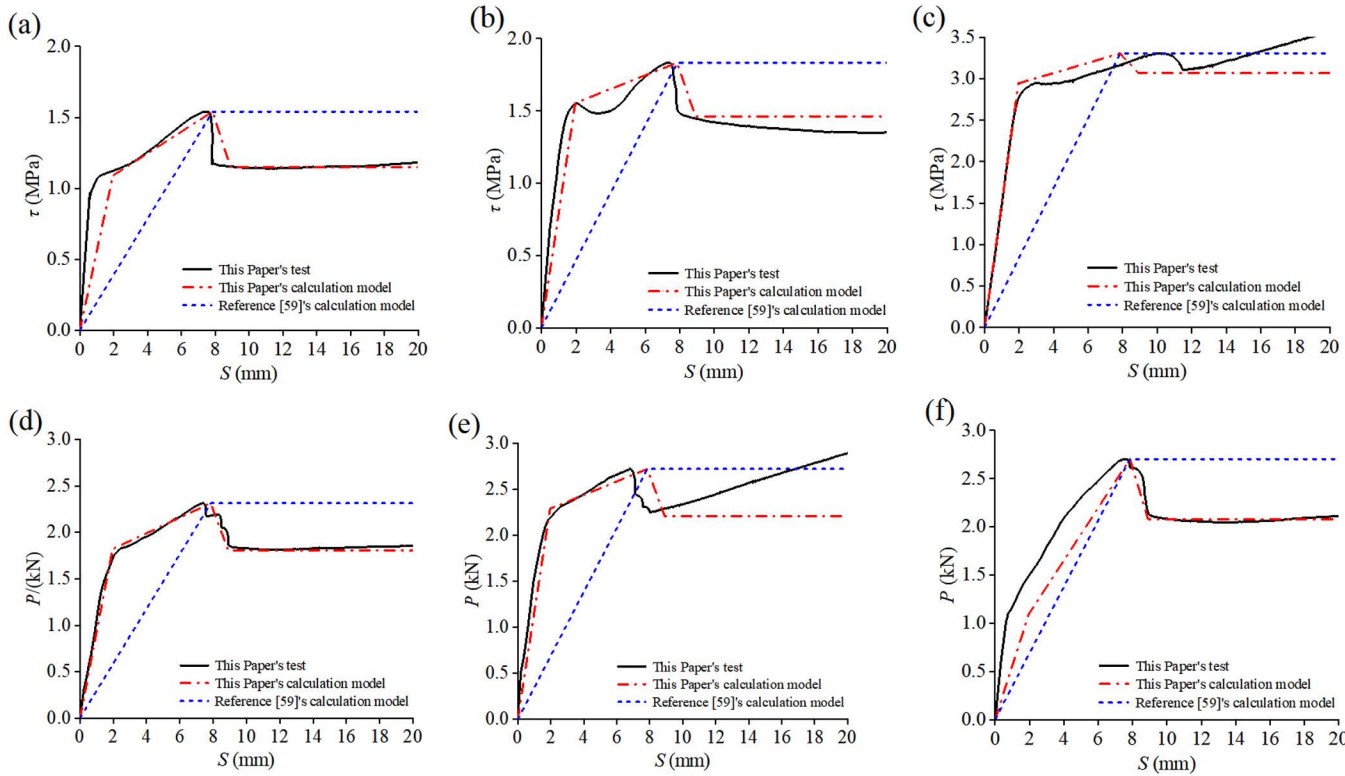

**Fig 16. Results of individual built-in studs specimens from tests, This Paper's calculation Model and Reference [58]'s calculation Model: (a) RACFST-2; (b) RACFST-3; (c) RACFST-4; (d) RACFST-5; (e) RACFST-6; (f) RACFST-7.**

**Table 5. Comparison Between This Paper's calculation Model and Reference [58]'s calculation Model on This Paper's Experimental Data Feature Points.**

| | Average deviation | RMSE | MAPE |
|---|---|---|---|
| This Paper | 0.187 | 0.231 | 11.32% |
| Reference [58] | 0.417 | 0.507 | 25.74% |

Dong conducted a push-out experiment on 15 concrete-filled steel tubes and found that the interfacial constitutive relationship should adopt the "two-stage formula" [58], as shown in Equation (9). The comparison results between the data and the constitutive model in this article and Equation (9) are shown in Fig 16. The quantitative comparison table of feature points is shown in Table 5.

$$\tau = \begin{cases} \frac{\tau_s}{S_s} S, & 0 \leq S \leq S_s \\ \tau_u & S_s \leq S \end{cases}$$

(9)

 Verify the formula with the bond-slip data from the reference [58], and the comparison results are shown in Fig 17.

 Compare the constitutive model of this study with the constitutive model in the reference [58]. The quantitative comparison table of feature points is shown in Table 6. It is found that the constitutive model of this study is in better agreement with the experimental result, which indicates that Equation (8) can more accurately describe the bond-slip characteristics of the bult-in studs RACFST.

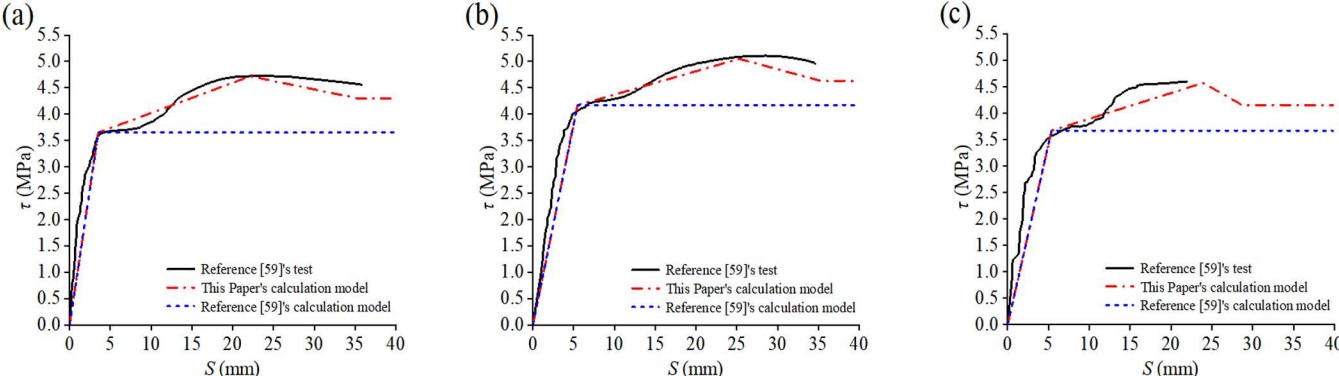

**Fig 17. Results of individual built-in studs specimens from tests of References [ 58], This Paper's calculation Model and Reference [58]'s calculation Model: (a) Specimen GZ1-C60; (b) Specimen GZ1-C70; (c) Specimen GZ1-RC60.**

**Table 6. Comparison Between This Paper's calculation Model and Reference [58]'s calculation Model on Reference [58]'s Experimental Data Feature Points.**

|  | Average deviation | RMSE | MAPE |
|---|---|---|---|
| This Paper | 0.089 | 0.107 | 2.06% |
| Reference [58] | 0.607 | 0.72308 | 14.65% |

## Conclusions

The position of the studs has a significant influence on the bonding performance of RACFST, the closer the stud is located to the free end, the greater the interfacial bond force becomes. The shear capacity of the RACFST interface with embedded studs is primarily provided by the natural bond between the steel tube and RAC, as well as the shear capacity of the studs. The longer the natural bond length that contributes to the shear resistance, the higher the ultimate bond force that can be achieved.

The number of studs also significantly impacts the bonding properties of RACFST. When the number of studs is small, the embedded studs may compromise the integrity of the interface, leading to a smaller interfacial bonding force. As the number of circumferential studs increases from 2 to 3, the ultimate bonding force rises by 42.48%; and as it further increases from 2 to 4 studs, the total increase in the ultimate bonding force is 76.70%. This enhancement in bonding force is stems from the increased combined stiffness of the RAC and studs, which overcomes any potential damage to the interface caused by the studs.

The stud promotes strain transfer the strain within RACFST. Upon installation, it facilitates an increased strain on the steel tube surface under identical load conditions, ensuring efficient transmission of the RAC's strain to the steel surface. This improves the overall performance and co-working capabilities of the composite structure. When the load is small, the stud's influence on the strain distribution is not significant, but the effect of the stud becomes evident when the load is large.

The new connection interfacial setting method proposed for numerical simulation addresses the deficiency of lacking a descending segment in traditional RACFST interfacial bond-slip simulation methods, and enables accurate simulation of the overall trend of the load-slip curve. The proposed bond-slip constitutive equation exhibits good agreement with experimental results, ensuring precise representation of the interfacial behavior, providing a basis for the application of RACFST with internal studs.

In summary, the quantity, position, and number of rows of shear studs also have a significant impact on the bonding performance of recycled aggregate concrete-filled steel tube (RACFST). They can redistribute the internal stress of RACFST and improve its overall working performance. A new interfacial setting procedure is proposed to overcome the previous limitation of being unable to fully simulate the bonding and sliding behaviors of the RACFST interface. This experimental study provides an experimental basis and theoretical foundation for RACFST, a practical and environmentally friendly material. Moreover, the new interfacial modeling method can be applied to more complex components or structures (e.g., steel-concrete frame structures, steel-concrete composite lattice members, etc.). In future research, we will further extend this bond-slip interfacial model to such complex structural systems.

## Supporting information

**S1 Data.** **Fig 1 Data : Coarse aggregate gradation data and fine aggregate gradation data. Fig 7 Data: The *P - S* data of the test specimens for the three series: Series of Stud Numbers, Series of Stud Positions, and Series of Stud Rows. Fig 8 Data : The longitudinal strain distribution data on the steel tube surface of the typical specimen. Fig 9 Data: The Transverse strain distribution data on the steel tube surface of the typical specimen. Fig 12 code : The code for "Model of bonding interface". Fig 14 Data : The comparative data of results from tests, simulations and calculations for individual built-in studs specimens. Fig 16 Data : The comparative τ - S data of results from tests, the calculation model in this paper, and the calculation model in Reference [58] for individual built-in studs specimens . Fig 17 Data : The comparative τ - S data of results from the tests in Reference [58], the calculation model in this paper, and the calculation model in Reference [58] for individual built-in studs specimens.**
(ZIP)

## Acknowledgments

The authors would like to thank the Civil Engineering Laboratory of University of South China for providing experimental conditions for this research.

## Author contributions

**Conceptualization:** Bing Sun.

**Formal analysis:** Gehao Cai.

**Methodology:** Bing Sun.

**Resources:** Jie Zhang.

**Validation:** Peng Yang.

**Writing – original draft:** Gehao Cai.

**Writing – review & editing:** Sheng Zeng.

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
