## [Decision Letter · Decision Letter 0]

Dear Dr. Sun,

Thank you for submitting your manuscript to PLOS ONE. After careful consideration, we feel that it has merit but does not fully meet PLOS ONE’s publication criteria as it currently stands. Therefore, we invite you to submit a revised version of the manuscript that addresses the points raised during the review process.

We look forward to receiving your revised manuscript.

Kind regards,

Dajiang Geng

Academic Editor

PLOS ONE

Journal Requirements:

“The research is supported by the Hunan Provincial Natural Science Foundation (No.2025JJ90163).”

“The research is supported by the Hunan Provincial Natural Science Foundation (No.2025JJ90163).”

“The research is supported by the Hunan Provincial Natural Science Foundation (No.2025JJ90163).”

6. In the online submission form, you indicated that “All relevant data are within the manuscript and its Supporting Information files. Some or all data, models, or code that support the findings of this study are available from the corresponding author upon reasonable request.”

7. When completing the data availability statement of the submission form, you indicated that you will make your data available on acceptance. We strongly recommend all authors decide on a data sharing plan before acceptance, as the process can be lengthy and hold up publication timelines. Please note that, though access restrictions are acceptable now, your entire data will need to be made freely accessible if your manuscript is accepted for publication. This policy applies to all data except where public deposition would breach compliance with the protocol approved by your research ethics board. If you are unable to adhere to our open data policy, please kindly revise your statement to explain your reasoning and we will seek the editor's input on an exemption. Please be assured that, once you have provided your new statement, the assessment of your exemption will not hold up the peer review process.

Reviewers' comments:

Reviewer's Responses to Questions

**Comments to the Author**

1. Is the manuscript technically sound, and do the data support the conclusions?

Reviewer #1: Yes

Reviewer #2: Yes

2. Has the statistical analysis been performed appropriately and rigorously?

Reviewer #1: Yes

Reviewer #2: Yes

3. Have the authors made all data underlying the findings in their manuscript fully available?

Reviewer #1: Yes

Reviewer #2: No

4. Is the manuscript presented in an intelligible fashion and written in standard English?

Reviewer #1: Yes

Reviewer #2: No

Reviewer #1: This manuscript explores the influence of built-in studs on the bond behavior within composite structures of recycled aggregate concrete- 14 filled steel tubes. Comments are provided as follows:

1. The quality of the figures needs to be improved. The reviewer can hardly see details of these figures.

2. Following references can be cited to improve the introduction section: A passive stress-strain model for concrete prisms reinforced by a combination of confinement reinforcement; Experimental study on ultimate bearing capacity of short thin-walled steel tubes reinforced with high-ductility concrete; Passive Stress-Strain Model and Ultimate Strength Prediction for Short Steel Tube Confined Concrete (STCC) Columns Allowing for Size Effect

3. On line 88, it is said that the concrete strength is C30. However, the average concrete strength should be tested and C30 is not concrete strength. Furthermore, the bond strength may also be affected by the average concrete strength which is not ‘C30’.

4. The reviewer thinks the section 4.1.3 for interfacial bonding is important and a flow chat is required to describe the model.

5. What is the conclusion for the section 4.3?

Reviewer #2: This study conducted push-out tests on seven different RACFST specimens with varying numbers, positions, and arrangements of bolt studs. The bonding slip mechanism at the RACFST interface was analyzed, and a corresponding constitutive relationship was established. A Python program was developed to set up the steel-concrete interface bonding model. While this research holds certain engineering significance, the paper presents the following shortcomings:

1. The language expression is somewhat unsatisfactory. For example, the Abstract repeatedly uses the first-person and active voice, and the Introduction contains numerous tense-related issues. The authors should carefully revise the entire manuscript to ensure it aligns more closely with academic conventions.

2. The core concrete material strength tests are missing. It is recommended that the authors provide additional testing details and data.

3. In Figure 10, there is a misspelling of "chemical"; this should be corrected.

4. How can the proposed interface bonding model be extended to the modeling of more complex components or structures (e.g., steel-concrete frame structures, steel-concrete composite lattice members, etc.)? Is it still possible to achieve accuracy and efficiency in such applications?

5. The "test curve" is misspelled in Figures 12, 14, and 15; it should be corrected.

6. In line 307, the specimen RACFST-4 is mentioned as being affected by macroscopic factors during the later stages of the test. Why did this phenomenon occur? How can it be avoided in actual engineering designs?

7. Although the paper mentions that the finite element model and the proposed constitutive model yield good prediction results, it does not provide a quantitative error analysis. Figures 12, 14, and 15 lack a quantitative comparison of characteristic points.

**Do you want your identity to be public for this peer review?** For information about this choice, including consent withdrawal, please see our Privacy Policy

Reviewer #1: No

Reviewer #2: No

---

## [Author Response · Author response to Decision Letter 1]

8 May 2025

Response Letter

Dear Editors and Reviewers.

We sincerely appreciate the time and effort taken by the Editors and Reviewers to evaluate our manuscript, "Investigation on the Bond-Slip Behavior of Recycled Aggregate Concrete-Filled Steel Tube with Studs". Their insightful comments and constructive suggestions have been instrumental in refining our work and enhancing the quality of the paper. We have carefully addressed each of the points raised, incorporating the recommended revisions into the manuscript. All modifications have been highlighted using the Track Changes feature in the revised version for ease of reference. Below, we provide a detailed response to the Editors’ and Reviewers’ comments, along with a summary of the key revisions made.

Detail Response to Reviewer 1

Dear reviewer,

Thank you for your letter and the reviewers’ comments concerning our manuscript. Those comments are valuable and very helpful. We appreciate the valuable comments. We have responded to your comments point by point and revised them one by one. In addition, we have marked the modified part in blue.

Comments & Reply:

1. The quality of the figures needs to be improved. The reviewer can hardly see details of these figures.

Reply:

Thanks to your suggestion. We have increased the resolution of all graphics and modified the curve labels in some of the images to make them clearer and more concise, in order to improve the visibility of these figures' details.

2. Following references can be cited to improve the introduction section: A passive stress-strain model for concrete prisms reinforced by a combination of confinement reinforcement; Experimental study on ultimate bearing capacity of short thin-walled steel tubes reinforced with high-ductility concrete; Passive Stress-Strain Model and Ultimate Strength Prediction for Short Steel Tube Confined Concrete (STCC) Columns Allowing for Size Effect

Reply:

Thanks for your suggestion. Upon careful review, we found the recommended references to be highly valuable for our study. References [1] and [2] demonstrate the confinement effect of steel tubes on the core recycled aggregate concrete (RAC), enhancing its ductility and strength, while also providing new insights into improving the bond strength of Recycled Aggregate Concrete-Filled Steel Tube�RACFST�. Additionally, as highlighted in Reference [3], bond strength plays a critical role in such composite structures, underscoring the significance of further research on enhancing the bond performance of RACFST. In light of these insights, we have revised the introduction section accordingly (see Page 2, Lines 36-40 in the revised manuscript) to incorporate these key references and strengthen the theoretical foundation of our work.

1.Hao XK, Zheng JJ, Fu CQ, Wang YT, Feng Q. Passive stress-strain model and ultimate strength prediction for short steel tube confined concrete (STCC) columns allowing for size effect. Case Stud Constr Mater, 2024;20: e02866.

2.Hao XK, Feng Q, Zheng JJ. A passive stress-strain model for concrete prisms reinforced by a combination of confinement reinforcement. Eng. Struct. 2021;246: 112981.

3.Chen RS, Zhang HY, Hao XK, Yu HX, Shi T, Zhou HS, et al. Experimental study on ultimate bearing capacity of short thin-walled steel tubes reinforced with high-ductility concrete. Structures. Elsevier. 2024;68: 107109.

3. On line 88, it is said that the concrete strength is C30. However, the average concrete strength should be tested and C30 is not concrete strength. Furthermore, the bond strength may also be affected by the average concrete strength which is not ‘C30’.

Reply:

Thanks for your suggestion, your suggestion is correct. In the article, "The concrete strength was C30" means that the concrete mix ratio was designed according to the strength grade C30. While we conducted compressive strength tests on the core concrete during the experimental phase, we inadvertently omitted these results in the original submission.To address this, we have now included detailed testing procedures as illustrated in Figure 1 and the test data in Table 1 (see Page 5, Lines 114–122 of the revised manuscript). These results further validate the material properties used in our study. We sincerely apologize for this oversight and thank the reviewer for bringing this important point to our attention, which has helped improve the completeness of our work.

Fig. 1 RAC compressive strength test.

Table. 2 Compressive strength of core RAC

Specimen RACFST-1 RACFST-2 RACFST-3 RACFST-4 RACFST-5 RACFST-6 RACFST-7

Compressive Strength(MPa) 34.1 32.5 33.0 31.8 33.3 32.7 32.5

4.The reviewer thinks the section 4.1.3 for interfacial bonding is important and a flow chat is required to describe the model.

Reply:

Thanks for your suggestion. Your suggestion is correct, a flowchart can make this section clearer and more straightforward. We have added a flowchart, as shown in Figure 2. We have added relevant content in the article, which can be found on page 11, lines 285-286.

Fig. 2 Flow chat of the new method for setting the connection interface

5.What is the conclusion for the section 4.3?

Reply:

Thanks for your suggestion. The conclusion of Section 4.3 is that after the installation of studs, the Mises stress distribution along the specimen height undergoes significant changes. The studs carry a substantial portion of the load, effectively reducing the bonding force sustained by the steel tube. This indicates that the studs modify the internal force transfer mechanism of the structure, resulting in a more uniform stress distribution, thereby improving the overall load-bearing capacity and structural stability. We have supplemented this discussion on Page 13, lines 341-344.

Detail Response to Reviewer 2

Dear reviewer,

Thank you for your letter and the reviewers’ comments concerning our manuscript. Those comments are valuable and very helpful. We have read through comments carefully and have made corrections. Based on the instructions provided in your letter, we uploaded the file of the revised manuscript. Revised portion are marked in blue in the paper.The responds to the reviewer’s comments are as flowing:

Comments & Reply:

1.The language expression is somewhat unsatisfactory. For example, the Abstract repeatedly uses the first-person and active voice, and the Introduction contains numerous tense-related issues. The authors should carefully revise the entire manuscript to ensure it aligns more closely with academic conventions.

Reply:

Thanks for your suggestion. We sincerely appreciate the reviewer's constructive suggestion. In response, we have implemented the following revisions: Revised the abstract (Page 2, lines 18-28), corrected verb tense usage throughout the introduction and conducted a comprehensive review and revision of the entire manuscript.

All modifications have been clearly marked in red font in the "Revised Manuscript with Track Changes" version to facilitate the reviewer's examination. These revisions have enhanced both the accuracy and readability of our manuscript. We are grateful for the reviewer's insightful comments, which have significantly improved the quality of our work.

2.The core concrete material strength tests are missing. It is recommended that the authors provide additional testing details and data.

Reply:

Thanks for your suggestion. While compressive strength tests on the core concrete were conducted during the experimental phase, we inadvertently omitted these results in the original submission. To address this, we have now added detailed testing procedures as illustrated in Figure 1, with all acquired data systematically presented in Table 1 (see Page 5, Lines 114–122 of the revised manuscript).

Fig. 1 RAC compressive strength test.

Table. 1 Compressive strength of core RAC

Specimen RACFST-1 RACFST-2 RACFST-3 RACFST-4 RACFST-5 RACFST-6 RACFST-7

Compressive Strength(MPa) 34.1 32.5 33.0 31.8 33.3 32.7 32.5

These additions further validate the material properties used in our study. We sincerely apologize for this oversight and appreciate the reviewer’s comment, which has improved the rigor and completeness of our work.

3.In Figure 10, there is a misspelling of "chemical"; this should be corrected.

Reply:

Thanks for your suggestion. The misspelling of “chemical” in Figure 10 has been corrected in the revised manuscript (see updated Figure 11).

4.How can the proposed interface bonding model be extended to the modeling of more complex components or structures (e.g., steel-concrete frame structures, steel-concrete composite lattice members, etc.)? Is it still possible to achieve accuracy and efficiency in such applications?

Reply:

Thanks for your suggestion. In this paper, we propose a novel interface modeling method to simulate the complete bond-slip behavior of RACFST with internal studs. This method can be directly applied to more complex components or structures (e.g., steel-concrete frame structures, steel-concrete composite lattice members, etc.) to establish their bond interface models. Regarding accuracy and efficiency, we agree that further validation is needed for such applications. Your insightful comment on extending the model to complex structural systems is highly appreciated and aligns with our planned future work. We will rigorously investigate these extensions in subsequent research, including numerical benchmarks and experimental comparisons to ensure both reliability and computational practicality.We have added relevant content in the article, which can be found on page 17, lines 430-434.

5.The "test curve" is misspelled in Figures 12, 14, and 15; it should be corrected.

Reply:

Thanks for your suggestion. The misspelling of “test curve” has been corrected in the revised manuscript (see updated figure 13, 16,and 17).

6.In line 307, the specimen RACFST-4 is mentioned as being affected by macroscopic factors during the later stages of the test. Why did this phenomenon occur? How can it be avoided in actual engineering designs?

Reply:

Thanks for your suggestion. Due to the irregular roundness of the steel tube or the inconsistent dimensions of its two ends, it may cause the curve to exhibit three development trends of ascending, descending, and stability, thus resulting in this macroscopic deviation. In practical engineering, this can be achieved through pre-installation inspection and correction, as well as optimizing construction processes (to avoid secondary deformation caused by external forces). We have added relevant content in the article, which can be found on page 13, lines 327-332.

7. Although the paper mentions that the finite element model and the proposed constitutive model yield good prediction results, it does not provide a quantitative error analysis. Figures 12, 14, and 15 lack a quantitative comparison of characteristic points.

Reply:

We sincerely appreciate the reviewer’s insightful suggestion regarding the need for quantitative validation of our models. To address this concern, we have taken the following steps in the revised manuscript:

(1)From Figure 12 (updated figure 13), it can be observed that the experimental data curve shows excellent agreement with the trends of both the finite element model and the proposed constitutive model. By comparing the experimental data with the ultimate bond strength results from the finite element model and the proposed constitutive model, we found that the finite element model maintains average errors within 7%, effectively simulating the entire bond-slip process. In contrast, the proposed constitutive model demonstrates extremely small errors (within 0.1%) in predicting the ultimate bond strength when compared with the experimental data. We have made corresponding revisions and additions in the relevant section of the paper (page 12, line 309-311 and page 14, line 361-363).

(2)The curve labels in Figures 14 and 15 (updated figure 16 and 17) have been modified to enhance clarity and ensure precise representation of the intended data. As evident from the plotted curves, the proposed computational model demonstrates improved agreement with experimental results compared to the reference model from [56]. Furthermore, a comparative analysis of key characteristic points was performed, with the corresponding error metrics between the two models systematically tabulated in Tables 2 and 3 for Figures 14 and 15 (page 16, line 396 and page 16, line 400-404).

Table. 2 Comparison Between This Paper's calculation Model and Reference [59]'s calculation Model on This Paper's Experimental Data Feature Points

Average deviation RMSE MAPE

This Paper 0.187 0.231 11.32%

Reference [59] 0.417 0.507 25.74%

Table. 3 Comparison Between This Paper's calculation Model and Reference [59]'s calculation Model on Reference [59]'s Experimental Data Feature Points

Average deviation RMSE MAPE

This Paper 0.089 0.107 2.06%

Reference [59] 0.607 0.72308 14.65%

---

## [Editor Report · Decision Letter 1]

Investigation on the Bond-Slip Behavior of Recycled Aggregate Concrete-Filled Steel Tube with Studs

PONE-D-25-21239R1

Dear Dr. Bing Sun,

We’re pleased to inform you that your manuscript has been judged scientifically suitable for publication and will be formally accepted for publication once it meets all outstanding technical requirements.

Kind regards,

Dajiang Geng

Academic Editor

PLOS ONE
---

## [Editor Report · Acceptance letter]

PONE-D-25-21239R1

PLOS ONE

Dear Dr. Sun,

I'm pleased to inform you that your manuscript has been deemed suitable for publication in PLOS ONE. Congratulations! Your manuscript is now being handed over to our production team.

Kind regards,

on behalf of

Dr. Dajiang Geng

Academic Editor

PLOS ONE